# Bayesian 3D X-ray Computed Tomography with a Hierarchical Prior Model for Sparsity in Haar Transform Domain

**DOI:** 10.3390/e20120977

**Published:** 2018-12-16

**Authors:** Li Wang, Ali Mohammad-Djafari, Nicolas Gac, Mircea Dumitru

**Affiliations:** Laboratoire des signaux et système, Centralesupelec, CNRS, 3 Rue Joliot Curie, 91192 Gif sur Yvette, France

**Keywords:** X-ray computed tomography, inverse problem, sparsity, hierarchical structure, generalized Student-*t* distribution, Haar transformation

## Abstract

In this paper, a hierarchical prior model based on the Haar transformation and an appropriate Bayesian computational method for X-ray CT reconstruction are presented. Given the piece-wise continuous property of the object, a multilevel Haar transformation is used to associate a sparse representation for the object. The sparse structure is enforced via a generalized Student-*t* distribution ($St_{g}$), expressed as the marginal of a normal-inverse Gamma distribution. The proposed model and corresponding algorithm are designed to adapt to specific 3D data sizes and to be used in both medical and industrial Non-Destructive Testing (NDT) applications. In the proposed Bayesian method, a hierarchical structured prior model is proposed, and the parameters are iteratively estimated. The initialization of the iterative algorithm uses the parameters of the prior distributions. A novel strategy for the initialization is presented and proven experimentally. We compare the proposed method with two state-of-the-art approaches, showing that our method has better reconstruction performance when fewer projections are considered and when projections are acquired from limited angles.

## 1. Introduction

Computed Tomography (CT) has been developed and widely used in medical diagnosis [1] and industrial Non-Destructive Testing (NDT) [2] in recent decades. In CT, objects are observed using different techniques, for example X-rays [3], ultrasound [4], microwaves [5], or infra-red [6]. X-ray CT employs the absorption of X-rays by the organs in a body or by the materials in industrial components to reconstruct the internal structure of the imaged object. When performing X-ray CT, a set of X-ray images of the measured parts is acquired. The intensity measured by the X-ray images corresponds to the intensity of the radiation passing through and attenuated by the object. CT reconstruction is typically treated as an inverse problem.

The conventional analytical techniques for CT reconstruction are based on the Radon transform [7]. Filtered Back-Projection (FBP) [8] is the most frequently-used analytical method in practical applications. FBP performs well when reconstructing from sufficient data with a high signal-to-noise ratio (SNR), but it suffers from artifacts when reconstructing from insufficient data or with noise.

Owing to considerations regarding patients’ health in medical CT and in order to reduce acquisition time in industrial applications, reconstruction with insufficient datasets is increasingly attracting the attention of researchers. Reconstruction from fewer projections is an ill-posed inverse problem [9,10]. In this case, conventional analytical reconstruction methods provide unsatisfactory results, and iterative methods can be used to improve the reconstruction performance. The Algebraic Reconstruction Technique (ART) [11,12], the Simultaneous Algebraic Reconstruction Technique (SART) [13], and the Simultaneous Iterative Reconstruction Technique (SIRT) [14,15] are some of the iterative methods proposed initially. These methods consider the discretized forward system model: g=Hf, where $f\in R^{N\times 1}$ represents the object, $g\in R^{M\times 1}$ represents the observed dataset, and matrix $H\in R^{M\times N}$ is the linear projection operator, mainly based on the geometry of acquisition (e.g., parallel beam, cone beam, etc.) [16,17,18]. Typically, the system of equations is under-determined, i.e., N>M. In this context, regularization methods are frequently used, and the forward system is modeled as: (1)g=Hf+ϵ,
where $\epsilon \in R^{M\times 1}$ represents the additive noise applied to the projection system. The regularization methods estimate the unknowns by minimizing a penalty criterion, which generally consists of two terms:(2)J(f)=Q(g,f)+λR(f).

The loss function Q(g,f) describes discrepancies in the observed data, such as the quadratic (L2) loss $Q\left(g,f\right)={∥g-Hf∥}_{2}^{2}$ or $L_{q}$ loss $Q\left(g,f\right)={∥g-Hf∥}_{q}^{q}$ with 1≤q<2. Other expressions such as the Huber function are also reported. The regularization term R(f) is a penalty on the complement criterion of f, such as restriction for smoothness ${∥\Phi (f)∥}_{2}^{2}$ or for sparsity ${∥\Phi (f)∥}_{1}$, where Φ(f) represents a linear function of f. The parameter λ is known as the regularization parameter, which controls the trade-off between the forward model discrepancy and the penalty term.

By choosing different regularization functions R(f), different regularization methods can be implemented. R(f)=0 refers to the Least-Squares (LS) method [19], with the drawback that the reconstruction is sensitive to the noise due to the ill-posedness of the problem and the ill-conditioning of the operator H. Quadratic Regularization (QR), also known as the Tikhonov method [20], is given by $R\left(f\right)={∥\Phi (f)∥}_{2}^{2}$, where the linear operator Φ(·) is the derivation operator in most cases. The well-known Total Variation (TV) method [21,22,23] is defined by $R\left(f\right)={∥D_{TV}f∥}_{TV}$ where $D_{TV}$ is the gradient operator. $D_{TV}$ is equal to ${∥D_{x}f∥}_{1}+{∥D_{y}f∥}_{1}+{∥D_{z}f∥}_{1}$ for a 3D object in an anisotropic form, where $D_{x}$, $D_{y}$ and $D_{z}$ are respectively the gradient operators in the *x*, *y* and *z* directions. The $L_{1}$ norm is used in TV for sparse estimations, which enforces the sparsity of $D_{TV}f$. The appearance of the non-differentiable $L_{1}$ term leads to difficulties for the implementation of optimization algorithms. Many optimization algorithms have been proposed to solve this $L_{1}$ norm optimization problem, for example the primal-dual method [24], the split Bregman method [22], etc. In regularization optimization, due to the large projection data size and the great number of voxels, the explicit expression of the solution cannot be used directly because of the impossibility of inversing the large size matrix such as ${\left(H^{T}H+\lambda D^{T}D\right)}^{-1}$. Hence, optimization algorithms such as gradient descent or conjugate gradient are often used.

More general regularization methods have been developed based on the constrained and dual-variable regularization method:(3)$$J\left(f,z\right)=Q_{1}\left(g,f\right)+\eta Q_{2}\left(f,z\right)+\lambda R\left(z\right),$$
which corresponds to the maximum a posterior optimization of a hierarchical structured model where both f and z are unknown variables. In such a model, the penalty regularization term is set on z, which is associated with f via a linear transformation. The loss functions $Q_{1}\left(g,f\right)$ and $Q_{2}\left(f,z\right)$ are for example quadratic (L2), i.e., $Q_{1}\left(g,f\right)={∥g-Hf∥}_{2}^{2}$ and $Q_{2}\left(f,z\right)={∥f-Dz∥}_{2}^{2}$ where D is a linear transform operator such as a wavelet transformation.

Among the methods treating this type of regularization problem, we mention here the Alternating Direction Method of Multipliers (ADMM) [25]. It minimizes Φ(f)+Ψ(z) subject to Af+Bz=C, and it covers a large number of estimation forms. One example is when $\Phi \left(f\right)={∥g-Hf∥}^{2}$, Ψ(z)=R(z), A=I, B=−D, and C=0 and refers to the above-mentioned bi-variable regularization method corresponding to Equation (Equation 3).

In the above-mentioned regularization methods, there is always a regularization parameter λ to be fixed. Sometimes, the regularization term consists of more than one parts, and each of them are weighted by a regularization parameter, for example the elastic-net regularizer [26]. In these cases, two or even more regularization parameters need to be fixed. Cross Validation (CV) and the L-curve method [20,27,28] are conventional methods used to determine suitable values for these parameters. However, this work must be repeated for different simulated datasets and is therefore very costly. Statistical methods, therefore, have been developed and used to solve this problem.

From the probabilistic point of view, a Gaussian model for the additive noise in the forward model, Equation (Equation 1), leads to the quadratic expression ${∥g-Hf∥}^{2}$ in the corresponding regularization criterion. However, in some types of tomography, for example Positron Emission Tomography (PET) or X-ray tomography with a very low number of phantoms, the noise is modeled by a Poisson distribution. In order to account for a more precise modeling of the noise and the other variables and parameters, statistical methods are used [29]. The Maximum Likelihood (ML) methods [30] and different estimation algorithms such as the Expectation Maximization (EM) algorithms [31], the Stochastic EM (SEM) [32], or the Ordered Subsets-EM (OS-EM) [33] are commonly used in PET-CT reconstruction problems.

Another widely-used type of probabilistic method for PET or X-ray CT reconstruction is the Bayesian inference [34,35,36,37]. The prior knowledge is translated by the prior probability model and is used to obtain the expression of the posterior distribution. The basic Bayesian formula is: (4)$$p\left(f|g,\theta \right)=\frac{p\left(g|f,\theta_{1}\right)p\left(f|\theta_{2}\right)}{p(g|\theta )},\;\mathrm{with}\;p\left(g|\theta \right)=\int p\left(g|f,\theta_{1}\right)p\left(f|\theta_{2}\right)\;df,$$
where $p(g|f,\theta_{1})$ is the likelihood, $p(f|\theta_{2})$ is the prior distribution, p(f|g,θ) is the posterior distribution, $\theta =\left(\theta_{1},\theta_{2}\right)$ are the parameters of these different distributions, and $p\left(g|\theta \right)$ is the evidence of the parameters in the data g. By using Maximum A Posterior (MAP) estimator $\hat{f}={\arg\max}_{f}\left\{p(f|g,\theta )\right\}={\arg\min}_{f}\left\{-\ln p(f|g,\theta )\right\}$, links between the Bayesian method and almost all the regularization methods can be illustrated. A Gaussian prior for p(f) in Equation (Equation 4) leads to the quadratic (L2) regularization method, while a Laplacian prior in Equation (Equation 4) leads to the L1 (LASSO or TV) regularization method. The regularization parameter can be related to $\theta_{1}$ and $\theta_{2}$. One advantage of the Bayesian method is having some explanation for the regularization parameter via its link with $\theta_{1}$ and $\theta_{2}$. For example, when $p(g|f,\theta_{1})$ and $p(f|\theta_{2})$ are Gaussian with $\theta_{1}$ and $\theta_{2}$, respectively the variances of the noise and the variance of the prior, then the regularization parameter is $\lambda =\theta_{1}/\theta_{2}$. Another advantage of the Bayesian method is that these parameters can also be estimated to achieve unsupervised or semi-supervised methods. This is achieved by obtaining the expression of the joint posterior probability law: (5)$$p\left(f,\theta |g\right)=\frac{p\left(g|f,\theta_{1}\right)p\left(f|\theta_{2}\right)p\left(\theta \right)}{p(g)},$$
where p(θ) is an appropriate prior on θ. For a hierarchical structured model where a hidden variable z appears in the prior model, we have: (6)$$p\left(f,z,\theta |g\right)=\frac{p\left(g|f,\theta_{1}\right)p\left(f|z,\theta_{2}\right)p\left(z|\theta_{3}\right)p\left(\theta \right)}{p(g)},$$
where $\theta =\left[\theta_{1},\theta_{2},\theta_{3}\right]$.

With the posterior distribution obtained from an unsupervised Bayesian inference as in Equation (Equation 5), we distinguish three estimation methods. The first method consists of integrating out θ from p(f,θ|g) to obtain p(f|g) and then using p(f|g) to infer on f. The second approach is firstly to marginalize p(f,θ|g) with respect to f to obtain p(θ|g)=∫p(f,θ|g)df and estimate $\hat{\theta }={\arg\max}_{\theta }\left\{p(\theta |g)\right\}$, then use $\hat{\theta }$ as it was known. Unfortunately, these approaches do not often give explicit expressions for p(f|g) or p(θ|g). The third and easiest algorithm to implement is the joint optimization, which estimates variable f and parameter θ iteratively and alternately. Bayesian point estimators such as Joint Maximum A Posteriori (JMAP) [38] and Posterior Mean (PM) [39] via Variational Bayesian Approximation (VBA) methods [40,41,42] are often used.

In order to distinguish the details of a reconstructed object, a high-resolution image is expected. In industrial applications, especially for the NDT of a large-size object, the size of the projection (1000 images of $1000^{2}$ pixels) and the number of voxels ($1000^{3}$ voxels) become critical, and so does the projection and back-projection operators in CT. It is necessary to account for some computational aspects, for example the GPU processor [43,44].

In our previous work [45], we proposed to use the Bayesian method via a synthesis model, in which the multilevel Haar transformation coefficient z of the image is first estimated, and then, the final image reconstruction result is obtained from post-processing: $\hat{f}=D\hat{z}$. In this case, when using a Laplacian prior model and the MAP estimator, the problem becomes equivalent to the optimization of $J\left(z\right)={∥g-HDz∥}^{2}+\lambda {∥z∥}_{1}$, which is a typical $L_{1}$ regularization method. The particularity of our work was to use a generalized Student-*t* ($St_{g}$) prior model [46] in place of the Laplacian model.

In this paper, we present a Hierarchical Haar transform-based Bayesian Method (HHBM), first proposed in [47], in which the object to be reconstructed, f, is related to the Haar transformation coefficient z by f=Dz+ξ where ξ represents the modelization uncertainties. f and z are estimated simultaneously. Wavelets provide an optimal representation for a piecewise continuous function consisting of homogeneous blocs separated by jump discontinuities (the contours), as the wavelet representation is sparse for such signals. The transformations used are, for example, the Haar transformation [48], the Curvelet Transformation (CVT) [49], the Contourlet Transformation (CT) [50], Dual-Tree Complex Wavelet Transform (DT-CWT) [51], etc. As long as the object under consideration, f, is piecewise continuous or constant, the Haar transform is appropriate, with the advantage that: (1) the transform coefficients are sparse; (2) the transformation operator is orthogonal so that the inverse operator and the transpose are identical; and (3) the computation of this transformation consists of only additions and subtractions, while the cost of computation is only $O(\sqrt{N})$ where *N* is the size of the object f.

The sparsity of the transformation coefficient is generally defined by three classes of distributions: the generalized Gaussian distributions [52], the mixture distributions [53], and the heavy-tailed distributions [54]. In this paper, we use a generalization of the Student-*t* distribution ($St_{g}$), which belongs to the heavy-tailed family and has many advantages when enforcing the sparsity of variables [46].

In this paper, we extend extensively the previous work by: (1) adapting the forward model and prior models to the 3D case, which is more appropriate for real 3D large data size applications; (2) comparing the RMSE of the phantom reconstructed by the HHBM method with those by the conventional QR and TV methods, we show the advantages of the semi-supervised property of the HHBM method and that the HHBM method outperforms the TV method when insufficient data are estimated; (3) proposing new ideas for fixing the hyper-parameters in the proposed model; and (4) evaluating the performance of the proposed method in the situations when the number or the angle distribution of the projections is limited.

The rest of this paper is organized as follows: Section 2 presents the proposed hierarchically-structured Bayesian method; Section 3 gives the details of the implementations and the choice of hyperparameters, as well as the simulation results; some points on the initialization of hyper-parameters are discussed in Section 4. Conclusions are drawn and prospective future research is presented in Section 5.

## 2. The Semi-Supervised Hierarchical Model

The Hierarchical Haar-based Bayesian Method (HHBM) is presented, in which the object f and its multilevel Haar transform coefficient z are considered jointly. A sparse enforcing prior is defined on z. The wavelet transformation has been used for the reconstruction of tomography images in some state-of-the-art works [55,56,57,58], using both regularization and Bayesian methods. In these state-of-the-art methods, the phantom f is obtained by a post-processing from reconstructed coefficient z. In this paper, the phantom f and the coefficient z are simultaneously estimated.

### 2.1. Forward System Model and Likelihood

In the proposed method, the forward model introduced in Equation (Equation 1) is considered. Generally, the noise in tomography is modeled by a Poisson distribution [59], but in X-ray CT, the Gaussian approximation is often used. We adopt the Gaussian approximation and propose to use a zero mean and non-stationary model where the variance is considered to be unknown, belonging to an inverse Gamma distribution given that this distribution provides a good adaptation of the positivity property of the variances $v_{\epsilon_{i}}$:
(7)$$\begin{array}{c} p\left(\epsilon |v_{\epsilon }\right)=N\left(\epsilon |0,V_{\epsilon }\right),\;V_{\epsilon }=\mathrm{diag}\left[v_{\epsilon }\right],\;\;\mathrm{where}\;\;v_{\epsilon }={\left[v_{\epsilon_{1}},\;\cdots ,\;v_{\epsilon_{M}}\right]}^{\prime }\in R^{M\times 1} \end{array}$$
(8)$$\begin{array}{c} p\left(v_{\epsilon }|\alpha_{\epsilon_{0}},\beta_{\epsilon_{0}}\right)=\prod_{i=1}^{M}IG\left(v_{\epsilon_{i}}|\alpha_{\epsilon_{0}},\beta_{\epsilon_{0}}\right) \end{array}$$
The vector $v_{\epsilon }$ is considered in order to account for the difference of sensitivity to noise for each detector in each projection direction.

According to the forward model of the linear system, Equation (Equation 1), and the prior model of the noise, Equation (Equation 7), the likelihood of this model system is: (9)$$p\left(g|f,v_{\epsilon }\right)=N\left(g|Hf,V_{\epsilon }\right).$$

In Bayesian inference, the likelihood is combined with the prior distributions to determine the posterior distribution.

### 2.2. Hierarchical Prior Model and Prior Distributions

Typically, the objects considered in medical and industrial X-ray CT are piecewise continuous. In this paper, a hierarchical prior model is used to define the piecewise continuous property. In this hierarchical prior model, a sparsity enforcing model is defined for the wavelet transformation coefficients of the image. A large number of methods accounting for the sparse structure of the solution have been proposed in the literature. Among them, the L1 regularization method is most frequently used, which minimizes the criterion $J\left(f\right)={∥g-Hf∥}_{2}^{2}+\lambda {∥\Phi f∥}_{1}$ where Φ is a linear operator, for example the gradient in the TV method. Another class of methods, known as “synthesis” [60], minimizes $J\left(z\right)={∥g-HDz∥}_{2}^{2}+\lambda {∥z∥}_{1}$ where f=Dz, and $z=D^{-1}f$ is for example a wavelet transform.

In this paper, we propose to use the multilevel Haar transformation as the sparse dictionary. The transformation is modeled using a discretized forward model: (10)f=Dz+ξ,
where $D\in R^{N\times N}$ represents the inverse multilevel Haar transformation and $\xi \in R^{N\times 1}$ represents the uncertainties of the transformation, which is introduced to relax the exact relation of the transform operator D. ξ is supposed to be sparse. Unlike the gradient operator used in the TV method, the multilevel Haar transformation is orthogonal, i.e., $D^{-1}=D^{T}$. This property provides certain advantages during optimization, especially for the big data size problems, as the inversion and transpose of the operator are identical and can be replaced by each other for different types of D.

ξ is modeled by a Gaussian distribution, $p\left(\xi \right)=N\left(\xi |0,V_{\xi }\right)$, where $V_{\xi }=\mathrm{diag}\left[v_{\xi }\right]$ and $v_{\xi }={\left[v_{\xi_{1}},\;\cdots ,\;v_{\xi_{N}}\right]}^{\prime }$. $v_{\xi }$ is considered an unknown variance. It is modeled in order to realize a semi-supervised system where the variance is estimated. Here, $v_{\xi }$ is modeled by an inverse Gamma distribution, with the same consideration as the model of $v_{\epsilon }$. The Gaussian model with an inverse Gamma distributed variance, $p\left(v_{\xi }|\alpha_{\xi_{0}},\beta_{\xi_{0}}\right)=\prod_{j=1}^{N}IG\left(v_{\xi_{j}}|\alpha_{\xi_{0}},\beta_{\xi_{0}}\right)$, leads to a generalized Student-*t* ($St_{g}$) distribution [46]. Consequently, a $St_{g}$ distribution is derived for ξ, and the sparse property of ξ can be guaranteed. From Equation (Equation 10), the conditional distribution $p(f|z,v_{\xi })$ is derived: (11)$$p\left(f|z,v_{\xi }\right)=N\left(f|Dz,V_{\xi }\right),$$
with: (12)$$p\left(v_{\xi }|\alpha_{\xi_{0}},\beta_{\xi_{0}}\right)=\prod_{j}^{N}IG\left(v_{\xi_{j}}|\alpha_{\xi_{0}},\beta_{\xi_{0}}\right).$$

For practical applications where these parameters are not known or difficult to obtain, we use a semi-supervised method in which the variances of noises, $v_{\epsilon }$ and $v_{\xi }$, are unknown. In HHBM, the inverse Gamma distribution is used to model $v_{\epsilon }$ and $v_{\xi }$, $p\left(v_{\epsilon }\right)=IG\left(v_{\epsilon }|\alpha_{\epsilon_{0}},\beta_{\epsilon_{0}}\right)$ and $p\left(v_{\xi }\right)=IG\left(v_{\xi }|\alpha_{\xi_{0}},\beta_{\xi_{0}}\right)$. Consequently, both ξ and ϵ are modeled by a $St_{g}$ distribution.

Vector $z={\left[z_{1},\;\cdots ,\;z_{N}\right]}^{\prime }$ represents the multilevel Haar transformation coefficient of piece-wise continuous f. As mentioned above, z is sparse. In this paper, the generalized Student-*t* distribution ($St_{g}$) [46] is used to enforce the sparsity structure of z. The $St_{g}$ distribution can be expressed as the marginal of a bivariate normal-inverse Gamma distribution: (13)$$St_{g}\left(z|\alpha ,\beta \right)=\int N\left(z|0,v\right)\;IG\left(v|\alpha ,\beta \right)\;\;dv.$$

Thanks to the fact that normal and inverse Gamma are conjugate distributions, the use of the $St_{g}$ via Equation (Equation 13) simplifies the computations when using the Bayesian point estimators such as the posterior mean via the Variational Bayesian Approximation (VBA) method [42].

From Equation (Equation 13), the $St_{g}$ prior distribution modeling z is expressed as the following model: (14)$$\begin{array}{c} p\left(z|v_{z}\right)=N\left(z|0,V_{z}\right),\;\;\mathrm{where}\;V_{z}=\mathrm{diag}\left[v_{z}\right],\;\;v_{z}=\left[v_{z_{1}},v_{z_{2}},\cdots ,v_{z_{N}}\right] \end{array}$$
(15)$$\begin{array}{c} p\left(v_{z}|\alpha_{z_{0}},\beta_{z_{0}}\right)=\prod_{j}^{N}IG\left(v_{z_{j}}|\alpha_{z_{0}},\beta_{z_{0}}\right), \end{array}$$
where $v_{z_{j}},\;\forall \;j=1:N$ are i.i.d. distributed. The difference between the standard St distribution, $St\left(z|\nu \right)=\int N\left(z|0,v\right)IG\left(v|\frac{\nu }{2},\frac{\nu }{2}\right)\;dv$, and the generalized $St_{g}$, given in Equation (Equation 13), is that St(z|ν) is governed by one parameter ν, but $St_{g}\left(z|\alpha ,\beta \right)$ is governed by two parameters (α,β). With these two parameters, the $St_{g}$ does not only enforce the sparsity of the variable, but also controls the sparsity rate [46]. By changing the values of the two hyper-parameters $\alpha_{z_{0}}$ and $\beta_{z_{0}}$, we can obtain either a heavy-tailed distribution with a narrow peak or a distribution approaching a Gaussian distribution.

### 2.3. The HHBM Method

The prior models of the proposed Bayesian method based on the forward model of Equation (Equation 1) and the prior model of Equation (Equation 10) are:(16)$$\begin{array}{c} p\left(g|f,v_{\epsilon }\right)\propto {\left|V_{\epsilon }\right|}^{-\frac{1}{2}}\exp\left[-\frac{1}{2}{\left(g-Hf\right)}^{T}V_{\epsilon }^{-1}\left(g-Hf\right)\right], \end{array}$$
(17)$$\begin{array}{c} p\left(f|z,v_{\xi }\right)\propto {\left|V_{\xi }\right|}^{-\frac{1}{2}}\exp\left[-\frac{1}{2}{\left(f-Dz\right)}^{T}V_{\xi }^{-1}\left(f-Dz\right)\right], \end{array}$$
(18)$$\begin{array}{c} p\left(z|v_{z}\right)\propto {\left|V_{z}\right|}^{-\frac{1}{2}}\exp\left[-\frac{1}{2}z^{T}V_{z}^{-1}z\right], \end{array}$$
(19)$$\begin{array}{c} p\left(v_{z}|\alpha_{z_{0}},\beta_{z_{0}}\right)\propto \prod_{j}^{N}v_{z_{j}}^{-(\alpha_{z_{0}}+1)}\exp\left[-\beta_{z_{0}}v_{z_{j}}^{-1}\right], \end{array}$$
(20)$$\begin{array}{c} p\left(v_{\epsilon }|\alpha_{\epsilon_{0}},\beta_{\epsilon_{0}}\right)\propto \prod_{i}^{M}v_{\epsilon_{i}}^{-(\alpha_{\epsilon_{0}}+1)}\exp\left[-\beta_{\epsilon_{0}}v_{\epsilon_{i}}^{-1}\right], \end{array}$$
(21)$$\begin{array}{c} p\left(v_{\xi }|\alpha_{\xi_{0}},\beta_{\xi_{0}}\right)\propto \prod_{j}^{N}v_{\xi_{j}}^{-(\alpha_{\xi_{0}}+1)}\exp\left[-\beta_{\xi_{0}}v_{\xi_{j}}^{-1}\right]. \end{array}$$

Figure 1 shows the generative graph of the proposed model in which the hyperparameters in the rectangles need to be initialized:

Via the Bayes rule, Equation (Equation 6), the joint posterior distribution of all the unknowns given in the data is derived: (22)$$\begin{array}{cc} p & (f,z,v_{\epsilon },v_{\xi },v_{z}\left|g\right)=\frac{p(g,f,z,v_{\epsilon },v_{\xi },v_{z})}{p(g)}\\ & =\frac{p\left(g|f,v_{\epsilon }\right)p\left(f|z,v_{\xi }\right)p\left(z|v_{z}\right)p\left(v_{z}\right)p\left(v_{\epsilon }\right)p\left(v_{\xi }\right)}{p(g)}\\ & \propto p\left(g|f,v_{\epsilon }\right)p\left(f|z,v_{\xi }\right)p\left(z|v_{z}\right)p\left(v_{z}\right)p\left(v_{\epsilon }\right)p\left(v_{\xi }\right). \end{array}$$

Bayesian point estimators are often used for estimation via the a posteriori distribution. In this paper, we focus on the JMAP estimation, given that in the case of the large data size of the 3D object, the computational costs for the VBA algorithm is too expensive.

### 2.4. Joint Maximum a Posteriori Estimation

The negative logarithm of the posterior distribution is used as the criterion of optimization in order to simplify the exponential terms. The maximization of the posterior distribution becomes a minimization of the criterion: (23)$$\begin{array}{cc} \left(f,z,v_{z},v_{\epsilon },v_{\xi }\right) & =\arg\max\left\{p(f,z,v_{\epsilon },v_{\xi },v_{z}|g)\right\}\\ & =\arg\min\left\{-\ln p(f,z,v_{\epsilon },v_{\xi },v_{z}|g)\right\}\\ & =\arg\min J(f,z,v_{z},v_{\epsilon },v_{\xi }). \end{array}$$

We substitute the distribution formulas and obtain: (24)$$\begin{array}{c} J\left(f,z,v_{z},v_{\epsilon },v_{\xi }\right)=-\ln p(f,z,v_{\epsilon },v_{\xi },v_{z}|g)\\ =\frac{1}{2}\sum_{i}^{M}\ln v_{\epsilon_{i}}+\frac{1}{2}{\left(g-Hf\right)}^{T}V_{\epsilon }^{-1}\left(g-Hf\right)\\ +\frac{1}{2}\sum_{j}^{N}\ln v_{\xi_{j}}+\frac{1}{2}{\left(f-Dz\right)}^{T}V_{\xi }^{-1}\left(f-Dz\right)\\ +\frac{1}{2}\sum_{j}^{N}\ln v_{z_{j}}+\frac{1}{2}z^{T}V_{z}^{-1}z+\left(\alpha_{z_{0}}+1\right)\sum_{j}^{N}\ln v_{z_{j}}\\ +\beta_{z_{0}}\sum_{j}^{N}v_{z_{j}}^{-1}+\left(\alpha_{\epsilon_{0}}+1\right)\sum_{i}^{M}\ln v_{\epsilon_{i}}+\beta_{\epsilon_{0}}\sum_{i}^{M}v_{\epsilon_{i}}^{-1}\\ +\left(\alpha_{\xi_{0}}+1\right)\sum_{j}^{N}\ln v_{\xi_{j}}+\beta_{\xi_{0}}\sum_{j}^{N}v_{\xi_{j}}^{-1}. \end{array}$$

The unknown variables are determined by obtaining the expressions of the alternate minimization in Equation (Equation 24): (25)$$\begin{array}{c} \hat{f}={\left(H^{T}{\hat{V}}_{\epsilon }^{-1}H+{\hat{V}}_{\xi }^{-1}\right)}^{-1}\left(H^{t}{\hat{V}}_{\epsilon }^{-1}g+{\hat{V}}_{\xi }^{-1}D\hat{z}\right), \end{array}$$
(26)$$\begin{array}{c} \hat{z}={\left(D^{T}{\hat{V}}_{\xi }^{-1}D+{\hat{V}}_{z}^{-1}\right)}^{-1}D^{T}{\hat{V}}_{\xi }^{-1}\hat{f}, \end{array}$$
(27)$$\begin{array}{c} \hat{v_{z_{j}}}=\left(\beta_{z_{0}}+\frac{1}{2}{\hat{z}}_{j}^{2}\right)/\left(\alpha_{z_{0}}+3/2\right), \end{array}$$
(28)$$\begin{array}{c} \hat{v_{\epsilon_{i}}}=\left(\beta_{\epsilon_{0}}+\frac{1}{2}{\left(g_{i}-{\left[H\hat{f}\right]}_{i}\right)}^{2}\right)/\left(\alpha_{\epsilon_{0}}+3/2\right), \end{array}$$
(29)$$\begin{array}{c} \hat{v_{\xi_{j}}}=\left(\beta_{\xi_{0}}+\frac{1}{2}{\left({\hat{f}}_{j}-{\left[D\hat{z}\right]}_{j}\right)}^{2}\right)/\left(\alpha_{\xi_{0}}+3/2\right), \end{array}$$
∀i∈[1,M] and ∀j∈[1,N].

In 3D X-ray CT, the inversion of matrix ${\left(H^{T}{\hat{V}}_{\epsilon }^{-1}H+{\hat{V}}_{\xi }^{-1}\right)}^{-1}$ and ${\left(D^{T}{\hat{V}}_{\xi }^{-1}D+{\hat{V}}_{z}^{-1}\right)}^{-1}$ in Equations (Equation 25) and (26) is impossible due to the large data size. First-order optimization methods are generally used in this case. In this paper, we use the gradient descent algorithm: (30)$$\begin{array}{c} \mathrm{for}\;k=1\to I_{G}:\;{\hat{f}}^{\left(k+1\right)}={\hat{f}}^{\left(k\right)}-{\hat{\gamma }}_{f}^{\left(k\right)}\nabla J_{f}\left({\hat{f}}^{\left(k\right)}\right), \end{array}$$
(31)$$\begin{array}{c} \mathrm{for}\;k=1\to I_{G}:\;{\hat{z}}^{\left(k+1\right)}={\hat{z}}^{\left(k\right)}-{\hat{\gamma }}_{z}^{\left(k\right)}\nabla J_{z}\left({\hat{z}}^{\left(k\right)}\right), \end{array}$$
where $I_{G}$ is the number of iterations for the gradient descent estimation and $\nabla J_{f}\left(\cdot \right)$ and $\nabla J_{z}\left(\cdot \right)$ are the derivatives of the criterion (Equation 24) regarding f and z, respectively. ${\hat{\gamma }}_{f}\left(\cdot \right)$ and ${\hat{\gamma }}_{z}\left(\cdot \right)$ are the corresponding descent step lengths, which are obtained by using an optimized step length strategy [61]: (32)$$\begin{array}{c} \nabla J\left(f\right)=-H^{T}V_{\epsilon }^{-1}\left(g-Hf\right)+V_{\xi }^{-1}\left(f-Dz\right), \end{array}$$
(33)$$\begin{array}{c} \nabla J\left(z\right)=-D^{T}V_{\xi }^{-1}\left(f-Dz\right)+V_{z}^{-1}z, \end{array}$$
(34)$$\begin{array}{c} {\hat{\gamma }}_{f}^{\left(k\right)}=\frac{{∥\nabla J({\hat{f}}^{\left(k\right)})∥}^{2}}{{∥{\hat{Y}}_{\epsilon }H\nabla J\left({\hat{f}}^{\left(k\right)}\right)∥}^{2}+{∥{\hat{Y}}_{\xi }\nabla J\left({\hat{f}}^{\left(k\right)}\right)∥}^{2}}, \end{array}$$
(35)$$\begin{array}{c} {\hat{\gamma }}_{z}^{\left(k\right)}=\frac{{∥\nabla J({\hat{z}}^{\left(k\right)})∥}^{2}}{{∥{\hat{Y}}_{\xi }D\nabla J\left({\hat{z}}^{\left(k\right)}\right)∥}^{2}+{∥{\hat{Y}}_{z}\nabla J\left({\hat{z}}^{\left(k\right)}\right)∥}^{2}}, \end{array}$$
where $Y_{\epsilon }=V_{\epsilon }^{-\frac{1}{2}}$, $Y_{\xi }=V_{\xi }^{-\frac{1}{2}}$, and $Y_{z}=V_{z}^{-\frac{1}{2}}$.

The algorithm concerning the optimization of all the unknowns is given in Algorithm 1.

**Algorithm 1** The JMAP Algorithm for the HHBM Method
  1:Fix parameters $\alpha_{z_{0}}$, $\beta_{z_{0}}$, $\alpha_{\epsilon_{0}}$, $\beta_{\epsilon_{0}}$, $\alpha_{\xi_{0}}$, $\beta_{\xi_{0}}$, *l*  2:**Input:**   H,D,g  3:**Output:**   $\hat{f},\;\hat{z},\;{\hat{v}}_{z},\;{\hat{v}}_{\epsilon },\;{\hat{v}}_{\xi }$  4:**Initialization:**    5:$\hat{f}\leftarrow$ normalized FBP  6:
$\hat{z}\leftarrow l-\mathrm{level}\;\mathrm{Haar}\;\mathrm{transformation}\;\mathrm{of}\;\hat{f}$
  7:**for**k=1 to $I_{max}$
**do**  8:        ${\hat{f}}^{\left(0\right)}=\hat{f}$  9:        **for**
k=1 to $I_{G}$
**do**10:                Calculate $\nabla J({\hat{f}}^{\left(k-1\right)})$ according to Equation (Equation 32)11:                Update ${\hat{\gamma }}_{f}^{\left(k\right)}$ according to Equation (Equation 34)12:                Update ${\hat{f}}^{\left(k\right)}={\hat{f}}^{\left(k-1\right)}-{\hat{\gamma }}_{f}^{\left(k\right)}\nabla J\left({\hat{f}}^{\left(k-1\right)}\right)$13:        **end for**14:        $\hat{f}={\hat{f}}^{\left(I_{G}\right)}$15:        ${\hat{z}}^{\left(0\right)}=\hat{z}$16:        **for**
k=1 to $I_{G}$
**do**17:                Calculate $\nabla J({\hat{z}}^{\left(k-1\right)})$ according to Equation (Equation 33)18:                Update ${\hat{\gamma }}_{z}^{\left(k\right)}$ according to Equation (Equation 35)19:                Update ${\hat{z}}^{\left(k\right)}={\hat{z}}^{\left(k-1\right)}-{\hat{\gamma }}_{z}^{\left(k\right)}\nabla J\left({\hat{z}}^{\left(k-1\right)}\right)$20:        **end for**21:        $\hat{z}={\hat{z}}^{\left(I_{G}\right)}$22:        Optimize $v_{z}$ according to Equation (Equation 27)23:        Optimize $v_{\epsilon }$ according to Equation (Equation 28)24:        Optimize $v_{\xi }$ according to Equation (Equation 29)25:
**end for**



## 3. Initialization and Experimental Results

For the simulations, the 3D simulated “Shepp–Logan” phantom, shown in Figure 2 on the left, Figure 3 on the top, and the 3D real “head” object, shown in Figure 2 on the right, Figure 3 on the bottom, both of size $256^{3}$, are used as the objects of interest to compare the performance of the proposed method to the performance of the other state-of-the-art methods. Both the Shepp–Logan and head phantoms consist of several different homogeneous areas, so both are piecewise continuous. The voxel values of the original objects are normalized to [0,1]. The projection directions are uniformly distributed, and each projection consists of $256^{2}$ detectors corresponding to a $256^{2}$ sized image.

The proposed HHBM method is compared with the conventional Quadratic Regularization (QR) and Total Variation (TV) methods. For the QR method, the gradient descent algorithm is used for the 3D large data size problem. For the TV method, the split Bregman method [22] is used to solve the $L_{1}$ norm minimization problem.

To evaluate the proposed method and compare it with the state-of-the-art methods, four different metrics are used:the Relative Mean Squared Error (RMSE), RMSE = ${∥f-\hat{f}∥}^{2}/{∥f∥}^{2}$, which shows a relative error of the results;the Improvement of the Signal-to-Noise Ratio (ISNR), which measures the improvement during iterations;the Peak Signal-to-Noise-Ratio (PSNR), which presents the SNR relative to the peak data value;the Structural Similarity of IMage (SSIM) [62], which evaluates the quality of the result approaching human vision.

In 3D X-ray CT, the projection matrix H is very large and is not accessible. For the simulations, only the projection operator Hf and the back-projection operator $H^{T}g$ are used. Considering that the costly projection and back-projection operators are computed in every iteration, the GPU processor is used via the ASTRA toolbox [63] to accelerate the computation.

### 3.1. Initializations

The initialization for the variables f and z, as well as the hyperparameters $\alpha_{\epsilon_{0}}$, $\beta_{\epsilon_{0}}$, $\alpha_{\xi_{0}}$, $\beta_{\xi_{0}}$, $\alpha_{z_{0}}$, and $\beta_{z_{0}}$ are discussed in this section.

The reconstructed phantom obtained by using the Filtered Back Projection (FBP) method is considered to be initial value ${\hat{f}}_{ini}$. The initialization of coefficient ${\hat{z}}_{ini}$ is the multilevel Haar transformation of ${\hat{f}}_{ini}$: ${\hat{z}}_{ini}=D^{-1}{\hat{f}}_{ini}$. In this article, we choose the level of transformation such that z has a sparse structure. As shown in Figure 4, when the transform level is small, for example two levels, the coefficient z is not sparse; when the transform level is sufficiently large, the coefficient is sparse. In this paper, we set z as a five-level Haar transform coefficient.

The initialization for $\alpha_{z_{0}}$ and $\beta_{z_{0}}$ is based on the sparse structure of the variable z. In Figure 4, we can see that the sparsity rate depends on the rank of transform coefficient *r*, where $r\in \left[1,\;l+1\right]$. For example, when l=2, shown in Figure 4, the coefficient has three ranks, $r\in \left[1,\;2,\;3\right]$. The first rank r=1 corresponds to the low frequency components in the transform coefficient.

The variable z is modeled by a $St_{g}$ distribution, with variance equal to $\mathrm{Var}\left[z_{j}|\alpha_{z_{0}},\beta_{z_{0}}\right]=\beta_{z_{0}}/\left(\alpha_{z_{0}}-1\right),\;\forall j\in \left[0,N\right]$. In this article, we fix the value $\alpha_{z_{0}}=2.1$ in order to have $\mathrm{Var}\left[z_{j}|\alpha_{z_{0}}=2.1,\beta_{z_{0}}\right]\approx \beta_{z_{0}},\;\forall j\in \left[0,N\right]$. The sparsity rate of z is defined by initializing different values for $\beta_{z_{0}}$, with a sparser structure when $\beta_{z_{0}}$ is smaller. $\beta_{z_{0}}$ is initialized as $\beta_{z_{0}}=10^{-r+1}$. When l=5, we have r=[1,2,3,4,5,6], and the hyperparameters $\beta_{z_{0}}=\left[10^{0},10^{-1},10^{-2},10^{-3},10^{-4},10^{-5}\right]$, respectively.

The initialization for the hyperparameters, $\alpha_{\epsilon_{0}},\;\beta_{\epsilon_{0}},\;\alpha_{\xi_{0}}$, and $\beta_{\xi_{0}}$, is based on the prior models of the variances $v_{\epsilon }$ and $v_{\xi }$ we have chosen. In the proposed method, we consider the background of the generalized Student-*t* distribution, in which both ϵ and ξ are modeled by a Gaussian distribution with inverse Gamma distributed variance, i.e., the $St_{g}$ distribution according to Equation (Equation 13).

The noise ϵ depends on the SNR of the dataset. In order to exploit this information in the initialization, we express the biased dataset as the sum of uncontaminated dataset $g_{0}$ and the additive noise ϵ: (36)$$g=g_{0}+\epsilon .$$

As the noise ϵ and the uncontaminated data $g_{0}$ are supposed to be independent, we have: (37)$${∥g∥}^{2}={∥g_{0}∥}^{2}+{∥\epsilon ∥}^{2}.$$

The SNR of the dataset is: (38)$$\mathrm{SNR}=10\log\frac{{∥g_{0}∥}^{2}}{{∥\epsilon ∥}^{2}}=10\log\frac{{∥g∥}^{2}-{∥\epsilon ∥}^{2}}{{∥\epsilon ∥}^{2}}.$$

With $E\left[\epsilon \right]=0$, we have: (39)$$v_{\epsilon }=E\left[\epsilon^{2}\right]\approx \frac{{∥\epsilon ∥}^{2}}{M}=\frac{{∥g∥}^{2}}{M}\times \frac{1}{1+10^{SNR/10}}.$$

The mean of variance $v_{\epsilon }$ of the noise ϵ is $E\left[v_{\epsilon_{i}}|\alpha_{\epsilon_{0}},\beta_{\epsilon_{0}}\right]=\beta_{\epsilon_{0}}/\left(\alpha_{\epsilon_{0}}-1\right),\;\forall i\in \left[0,M\right]$, so we obtain:(40)$$\beta_{\epsilon_{0}}=\frac{{∥g∥}^{2}}{M}\times \frac{1}{1+10^{SNR/10}}\times \left(\alpha_{\epsilon_{0}}-1\right).$$

The two hyperparameters $\alpha_{\epsilon_{0}}$ and $\beta_{\epsilon_{0}}$ are combined according to Equation (Equation 40); hence, initialization for one of them is sufficient. In real applications, the SNR of the dataset is unknown, but we can use the projection of an empty object, i.e., f=0, to obtain a rough value of the variance of noise $v_{\epsilon }$.

Figure 5 shows the influence of the value of $\alpha_{\epsilon_{0}}$ on the reconstruction. According to the results, a bigger value for $\alpha_{\epsilon_{0}}$ results in a smaller value on RMSE for different numbers of projections and the SNR of the dataset. This monotonous property facilitates the initialization of this hyperparameter, as a large value for $\alpha_{\epsilon_{0}}$ satisfies all cases. When $\alpha_{\epsilon_{0}}$ is greater than a threshold value, the RMSE does not change with different initialization values for $\alpha_{\epsilon_{0}}$.

For ξ, both $\alpha_{\xi_{0}}$ and $\beta_{\xi_{0}}$ are analyzed for the influence of the reconstruction results.

Figure 6 shows the influence of the hyperparameter $\alpha_{\xi_{0}}$. Different colors represent an initialization with a different value of $\beta_{\xi_{0}}$. As here we focus on the analysis of $\alpha_{\xi_{0}}$ for all cases of $\beta_{\xi_{0}}$ values, we do not show the corresponding $\beta_{\xi_{0}}$ value for each different color. For different noise levels, different numbers of projections, and different $\beta_{\xi_{0}}$ values, the RMSE has an upward trend when the value of $\alpha_{\xi_{0}}$ becomes larger.

Figure 7 shows the influence of the hyperparameter $\beta_{\xi_{0}}$. Different colors represent an initialization with a different hyperparameter $\alpha_{\xi_{0}}$. For different noise levels, different numbers of projections, and different values for $\alpha_{\xi_{0}}$, when the value of $\beta_{\xi_{0}}$ increases, the RMSE decreases, then, after a slight increase, levels out.

In [46], it is pointed out that when α and β of the $St_{g}$ distribution are both large, the $St_{g}$ distribution approaches a Gaussian distribution, which is the case for the additive noise ϵ. If α and β are very small (approaching zero), the $St_{g}$ distribution becomes a non-informative distribution (Jeffreys distribution); when α and β are both small, the $St_{g}$ has the sparsity enforcing property, which is the case for the sparse ξ. Consequently, the initialization of the hyperparameters is theoretically supported, and they can be initialized with respect to these properties in other simulations.

### 3.2. Simulation Results with a Limited Number of Projections

We apply 180,90,60,45,36, and 18 projections evenly distributed in $\left[0,180\right]$ degrees for the reconstruction of the 3D Shepp–Logan phantom of size $256^{3}$; each projection contains $256^{2}$ detectors. The number of projections is chosen such that there is respectively one projection every 1,2,3,4,5, and 10 degrees.

In Table 1, different evaluation metrics of the reconstructed $256^{3}$ Shepp–Logan phantom are compared. It is shown that the HHBM method does not always perform better than the TV method, especially when there are sufficient numbers of projections. However, when there is insufficient projection data, the HHBM method is more robust than the TV method. On the other hand, as it is known that the choice of regularization parameter plays an important role in the regularization methods like QR or TV and the value for the regularization parameter should be selected for each different system settings, the HHBM method is much more robust on the initialization of hyper-parameters. As we can see from Figure 5, Figure 6 and Figure 7, once we have chosen the hyperparameters in a certain interval, which is not difficult to fix according to the properties of the prior model, we can obtain the appropriate reconstruction results. More importantly, in the Bayesian approach, the prior model can be chosen from a variety of other suitable distributions, which gives more possibilities for the models than the conventional regularization methods. We may also choose different point estimators from the posterior distribution, for example the posterior mean, etc.

Figure 8 and Figure 9 show the reconstructed middle slice of the “Shepp–Logan” phantom and “head” object by using the TV and HHBM methods from 36 projections with SNR = 40 dB and SNR = 20 dB. The red curve illustrates the profile of the blue line position. In the reconstructed Shepp–Logan phantom obtained using the TV method, the three small circles on the top of the slice are not evident. By using the HHBM method, we can distinguish these three small circles. By comparing the profiles of the slice of the reconstructed Shepp–Logan phantom, we can see that by using the HHBM method, the contour positions on the profile are closer to the original profile than those obtained using the TV method. In the reconstructed head object, there are more details than the simulated Shepp–Logan phantom, especially in the zoom area in the second line in Figure 8. By comparing the results, we can see that for the type of object that contains some small details, the TV method derives a result with smoother homogeneous areas, but with fewer details in the contour areas than the HHBM method. Some of the white material, which is dispersed into discontinuous small blocks in the head object, is connected in the results of the TV method. From these images, we conclude that with an insufficient number of projections, the proposed method gives results with clearer contours and details.

Figure 10 shows the reconstructed Shepp–Logan phantom from 18 projections of SNR = 40 dB and 20 dB. In this very underdetermined case, the HHBM method can still obtain a result that is clear enough to distinguish the primary zones and contours of the object.

Figure 11 and Figure 12 show the comparison between the QR, TV, and HHBM methods with a high SNR = 40 dB and a low SNR = 20 dB dataset, respectively. The abscissa corresponds to the number of projections evenly distributed from $0^{\circ }$–$180^{\circ }$, and the ordinate is the RMSE after 50 iterations. In the simulations, we used an SNR = 40 dB to represent a weak noise case and SNR = 20 dB for a strong noise case. When SNR = 40 dB, the HHBM method outperforms both quadratic regularization and the TV method. When SNR = 20 dB, the TV method with the optimal regularization parameter outperforms the HHBM method. However, in Figure 12, we show another curve in light green (TV2) showing the TV reconstruction with some random regularization parameters, which are chosen as a value not far from the optimal regularization parameters. From these results, we can see that the HHBM method is more robust than the TV reconstruction method with respect to the regularization parameter, the optimal value of which is, on the other hand, difficult to determine in the real applications where we cannot evaluate the estimation quality.

### 3.3. Simulation Results with a Limited Angle of Projections

In both medical and industrial X-ray CT, another common challenge is the limit of projection angles. In this part of the simulation, we use evenly-distributed projections in a limited range of angles for the simulated 3D “Shepp–Logan” phantom and the 3D “head” object, both of which have a size of $256^{3}$.

Figure 13 and Figure 14 show the middle slice of the reconstructed Shepp–Logan phantom and the head object, from 90 projections distributed between $0^{\circ }$ and $90^{\circ }$, with projection SNR of 40 dB and 20 dB. By using the TV method, the reconstructed object is blurry along the diagonal direction for which there is no projection data, and there is a square corner where the object should have a rounded edge. By using the HHBM method, we get results that are more consistent with the true shape and clearer contours.

Figure 15 and Figure 16 show the comparison of the performance in terms of RMSE of different methods with a high SNR = 40 dB and a low SNR = 20 dB. In this comparison, four cases of limited projection angles are considered, and they are: 45, 90, 135, and 180 projections evenly distributed in $\left[0^{\circ },45^{\circ }\right]$, $\left[0^{\circ },90^{\circ }\right]$, $\left[0^{\circ },135^{\circ }\right]$, and $\left[0^{\circ },180^{\circ }\right]$, respectively. There is one projection every $1^{\circ }$. From these two figures, we conclude that the proposed HHBM method remains more robust than the other two conventional methods when there are limited projection angles.

### 3.4. Simulation with a Different Forward Model

To study the robustness of the results with respect to the modeling errors, we apply a slightly different forward model for the one used for the generation of the simulated data and the one used for the reconstruction. During the projection, the projector is applied on the Shepp–Logan phantom of size $1024^{3}$, with the detector size of $256^{2}$ for each projection direction.

In Figure 17, we show the results of the two projection data obtained, as well as their difference $\delta_{proj}$, which can be considered as the forward modeling error. The $\delta_{proj}$ is rescaled in order to show clearly the details. We then use the data obtained from the $1024^{3}$ phantom to reconstruct an object of size $256^{3}$.

In Figure 18, the middle slices of the reconstructed Shepp–Logan phantom by using the QR, TV, and HHBM methods are presented, by using 180 and 36 projections, respectively. From the figures, we can see that when there are 180 projections, all three methods performs well, and the TV method detects better the contours, while the HHBM method has more noise at the contour areas. When there are insufficient projection numbers (36 projections in this simulation), the HHBM method outperforms the QR and TV methods for reconstructing the details in the phantom, for example the three small circles in the top of the phantom.

In Table 2, the RMSE of the reconstructed phantom by using the different methods are compared. We can conclude that, when the projector model is different than the reconstruction one, all these three methods (QR, TV, and HHBM) have good performance when there are 180 projections. When the projection number decreases, the TV and HHBM methods outperform the QR method. Comparing with the TV method, the HHBM method is more robust to the number of projections. When the projection number is smaller than 60, the HHBM method outperforms the TV method.

All the MATLAB codes for the simulations in this paper can be found on GitHub [64].

## 4. Discussion

One advantage of the Bayesian approach is the estimation of the parameters along with the estimation of unknown variables of the forward model at each iteration. However, like in regularization methods, the hyper-parameters need to be initialized.

While the parameters in the regularization methods play an important role in the final results and they are costly to fix, the hyper-parameters in HHBM can be initialized based on the prior information (the sparse structure of z) and the prior model (the Student-*t* distribution). In this article, we have shown that once the hyper-parameters are fixed in a certain appropriate interval, which is not difficult to obtain, the corresponding algorithm is robust. In this work, the hyper-parameters are not fixed via the classical approach using non-informative prior laws (i.e., considering the inverse Gamma corresponding parameters such that they approach Jeffreys’) [65].

## 5. Conclusions

In this paper, we propose a Bayesian method with a hierarchical structured prior model based on multilevel Haar transformation (HHBM) for 3D X-ray CT reconstruction. Simulation results indicate that for a limited number of projections or limited projection angles, the proposed method is more robust to noise and to regularization parameters than the classical QR and TV methods.

Indeed, we observe a relatively weak influence of the hyper-parameters in the behavior of the corresponding iterative algorithm. The interest of this weak dependency is that it offers a practical way to ensure the initialization of the algorithm, which typically is not-trivial.

In this context, as future work, we are investigating the causes of the relatively weak influence of the hyper-parameters and the theoretical foundation of the corresponding robust interval, extending the discussion to the same approach using sparsity-enforcing priors expressed as normal variance mixtures, but for other mixing distributions (Gamma, generalized inverse Gaussian) [66].

Another extension of this work is to consider the posterior mean as an estimator. This can be done via the Variational Bayesian Approach (VBA), but a practical implementation requires a method of accessing the diagonal elements of the large matrix $H^{T}H$, which is being studied by our group.

## Figures and Tables

**Figure 1 entropy-20-00977-f001:**
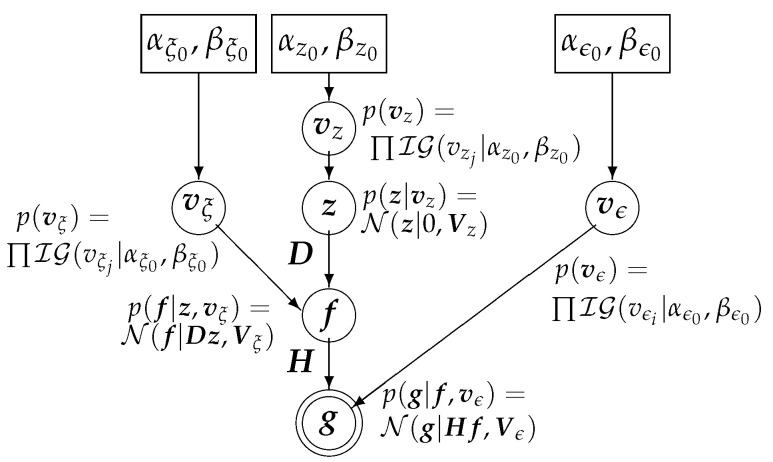
Generative graph of the proposed model illustrating all the unknowns (circles), hyperparameters (boxes), and data (double circles).

**Figure 2 entropy-20-00977-f002:**
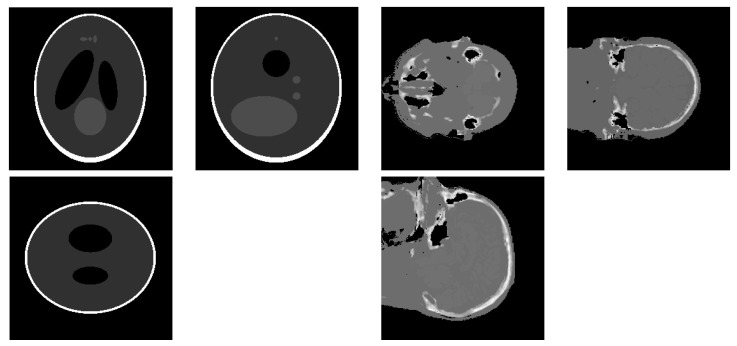
The three figure on the left show the three middle slice views of the three-dimensional Shepp–Logan phantom, and the three figures on the right show the three-dimensional head phantom.

**Figure 3 entropy-20-00977-f003:**
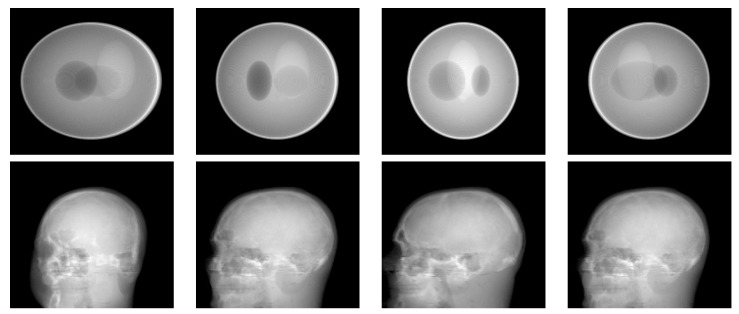
(**top**) Four projections of the 3D SheppLogan phantom from 30, 60, 90, 120 degrees (left to right, respectively); (**bottom**) four projections of the 3D head phantom from 30, 60, 90, 120 degrees (left to right, respectively).

**Figure 4 entropy-20-00977-f004:**
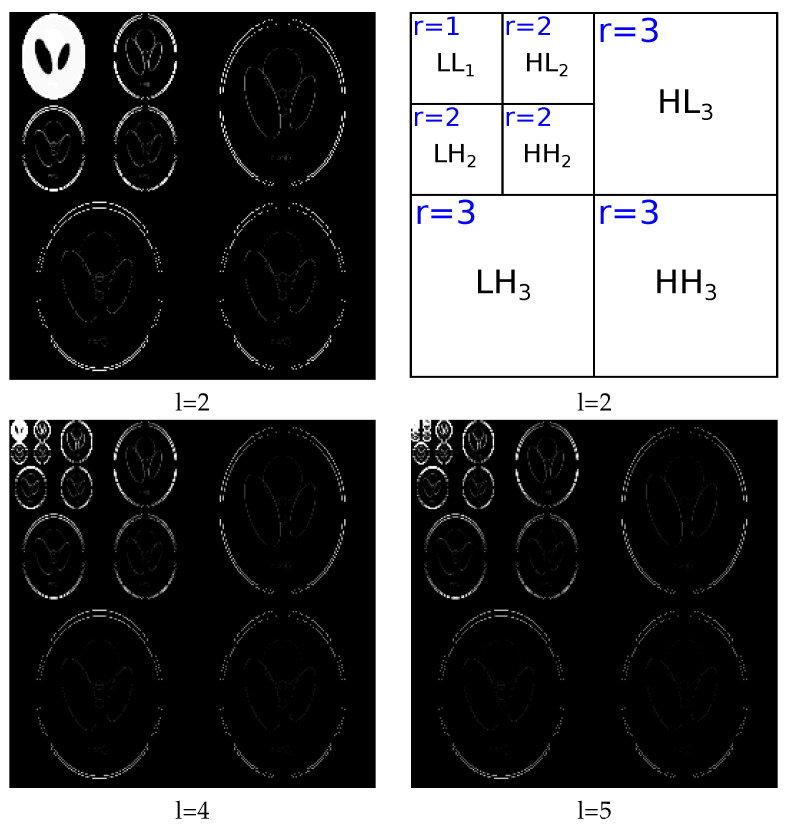
Slice of the 3D multilevel Haar transformation coefficient z of the $256^{3}$ Shepp–Logan phantom with: 2 levels (top-left), 4 levels (bottom-left), and 5 levels (bottom-right). The figure on the top-right shows the ranks of coefficients for a two-level transformation.

**Figure 5 entropy-20-00977-f005:**
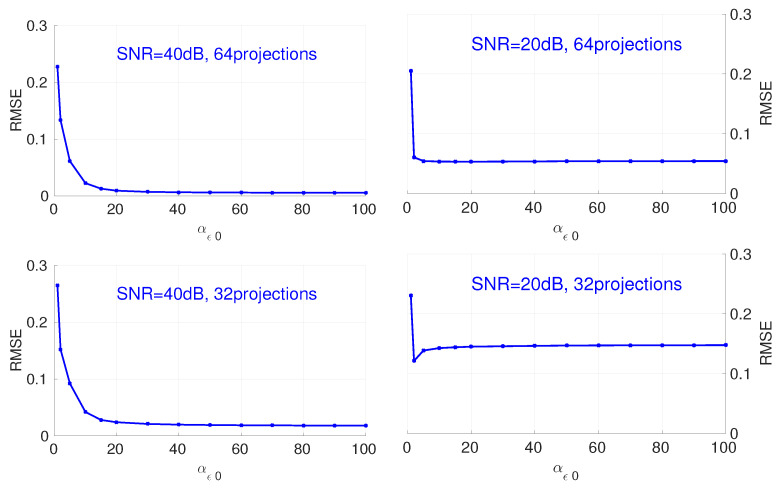
Influence of hyperparameter $\alpha_{\epsilon_{0}}$ on the RMSE of final reconstruction results for different numbers of projections and noise.

**Figure 6 entropy-20-00977-f006:**
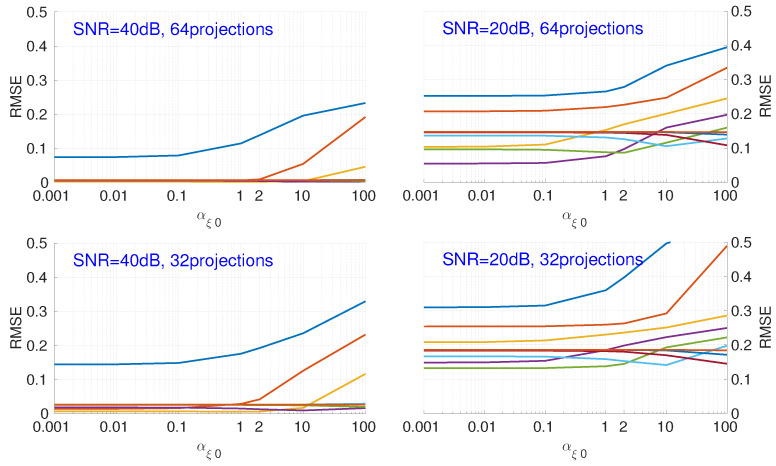
Influence of hyperparameter $\alpha_{\xi_{0}}$, with different fixed values of $\beta_{\xi_{0}}$, on the RMSE of reconstruction results for different numbers of projections and noise. Each color corresponds to a different initialization of $\beta_{\xi_{0}}$.

**Figure 7 entropy-20-00977-f007:**
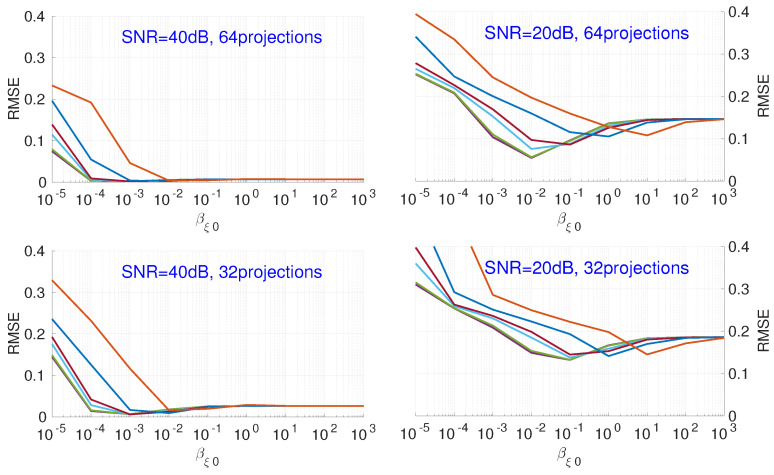
Influence of hyperparameter $\beta_{\xi_{0}}$, with different fixed values of $\alpha_{\xi_{0}}$, on the RMSE of reconstruction results for different numbers of projections and noise. Each color corresponds to a different initialization of $\alpha_{\xi_{0}}$.

**Figure 8 entropy-20-00977-f008:**
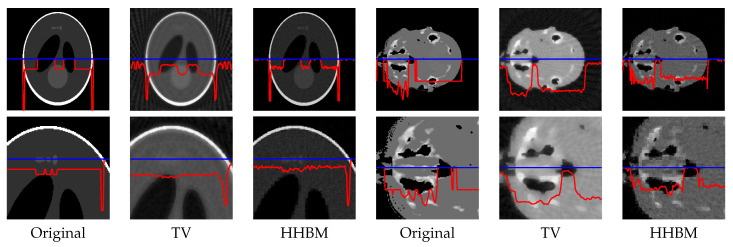
Reconstructed “Shepp–Logan” phantom and “head” object of size $256^{3}$, with a dataset of 36 projections and SNR = 40 dB, by using the TV and HHBM methods. The bottom figures are zones of the corresponding top figures. The red curves are the profiles at the position of the corresponding blue lines.

**Figure 9 entropy-20-00977-f009:**
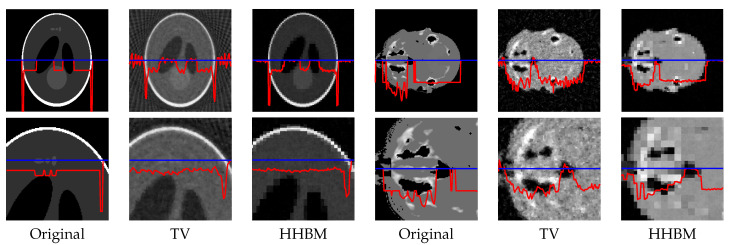
Reconstructed “Shepp–Logan” phantom and “head” object of size $256^{3}$, with a dataset of 36 projections and SNR = 20 dB, by using the TV and HHBM methods. Bottom figures are zones of the corresponding top figures. The red curves are the profiles at the position of the corresponding blue lines.

**Figure 10 entropy-20-00977-f010:**
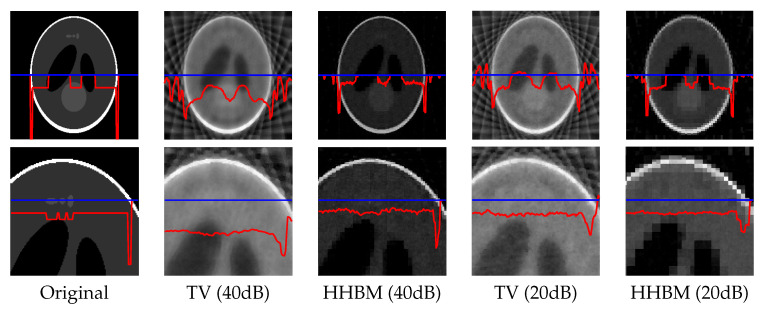
Reconstructed Shepp–Logan phantom of size $256^{3}$, with a dataset of 18 projections of SNR = 40 dB and 20 dB, by using the TV and HHBM methods. The red curves are the profiles at the position of the corresponding blue lines.

**Figure 11 entropy-20-00977-f011:**
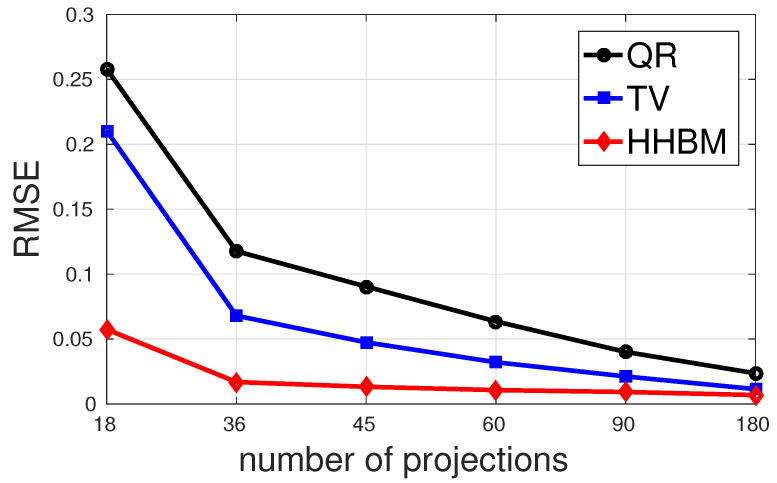
The performances of different methods for reconstructing the Shepp–Logan phantom in terms of RMSE with different numbers of projections evenly distributed in $\left[0^{\circ },180^{\circ }\right]$ and a high SNR = 40 dB.

**Figure 12 entropy-20-00977-f012:**
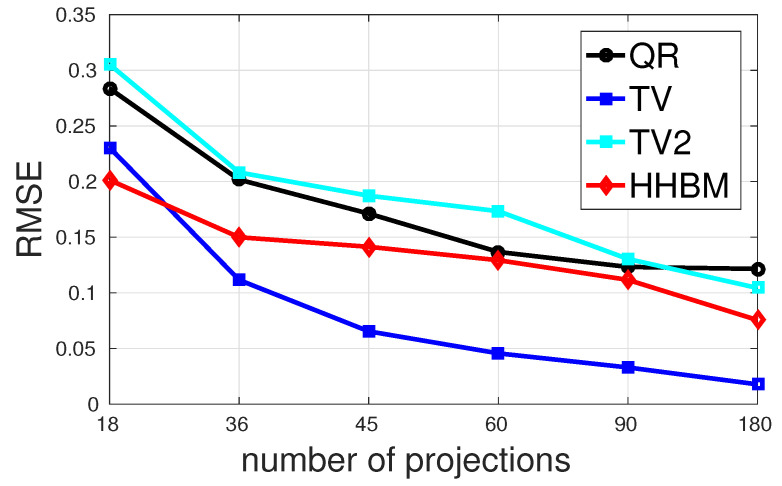
The performances of different methods for reconstructing the Shepp–Logan phantom in terms of RMSE with different numbers of projections evenly distributed in $\left[0^{\circ },180^{\circ }\right]$ and a low SNR = 20 dB.

**Figure 13 entropy-20-00977-f013:**
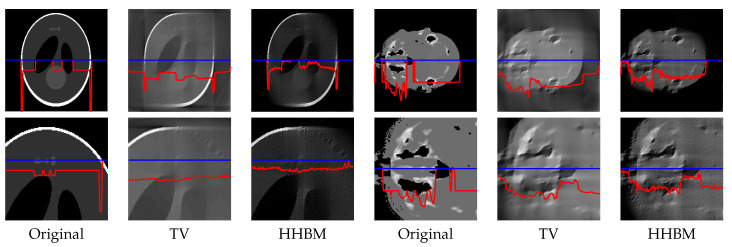
Slice of the reconstructed 3D Shepp–Logan phantom and 3D head object, with 90 projections evenly distributed in $\left[0^{\circ },90^{\circ }\right]$, SNR = 40 dB. The bottom figures are parts of the corresponding top figures. The red curves are the profiles at the position of the corresponding blue lines.

**Figure 14 entropy-20-00977-f014:**
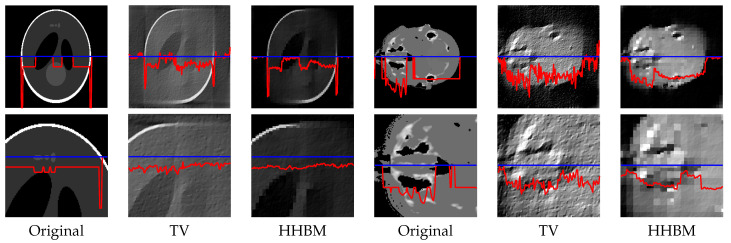
Slice of the reconstructed 3D Shepp–Logan phantom and 3D head object, with 90 projections evenly distributed in $\left[0^{\circ },90^{\circ }\right]$, SNR = 20 dB. Bottom figures are parts of the corresponding top figures. The red curves are the profiles at the position of the corresponding blue lines.

**Figure 15 entropy-20-00977-f015:**
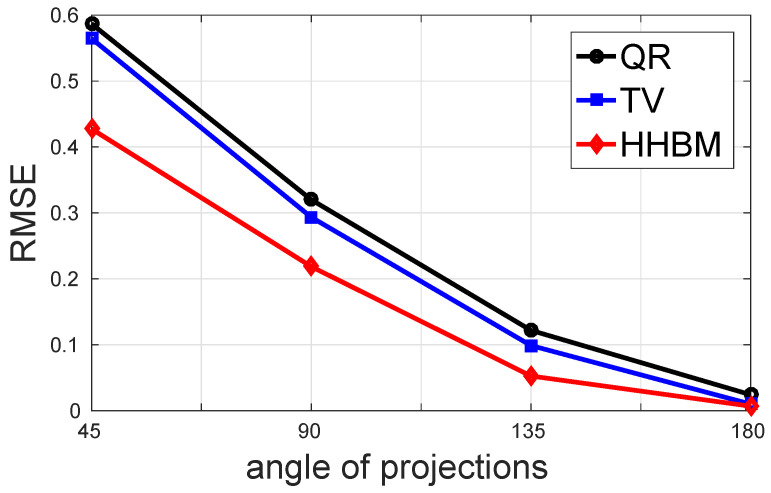
The performance of different methods for reconstructing the Shepp–Logan phantom in terms of RMSE with different limited projection angles and a high SNR = 40 dB.

**Figure 16 entropy-20-00977-f016:**
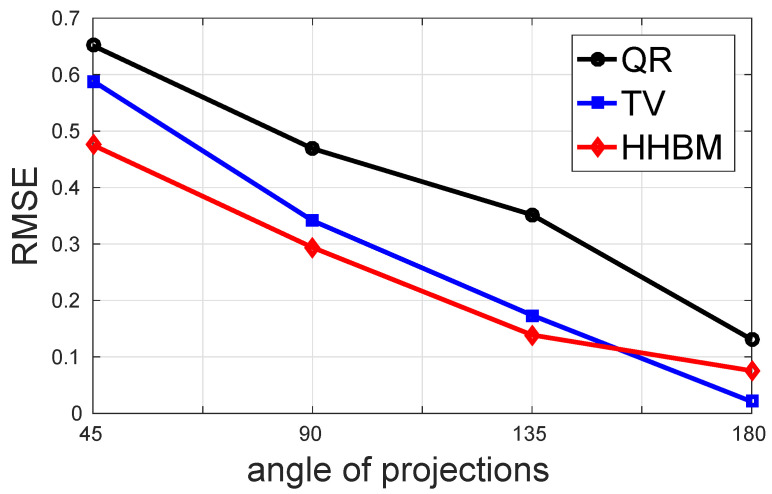
The performance of different methods for reconstructing the Shepp–Logan phantom in terms of RMSE with different limited projection angles and a low SNR = 20 dB.

**Figure 17 entropy-20-00977-f017:**
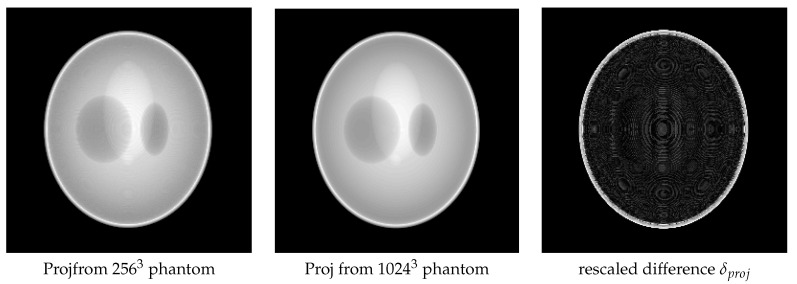
The projection image at an angle pf 90 degrees by using two different forward models: projection from the $256^{3}$ phantom (**left**) and projection from the $1024^{3}$ phantom (**middle**). The difference between them is shown on the right.

**Figure 18 entropy-20-00977-f018:**
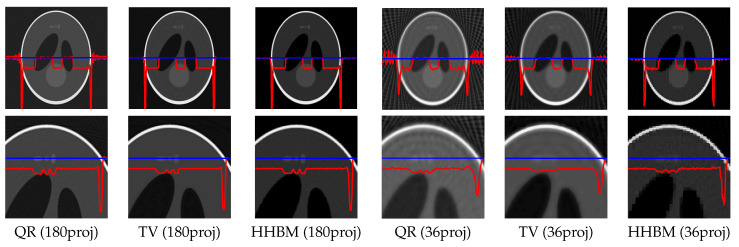
Reconstructed phantom with different forward models and 180 projections and 36 projections by using the QR (**left**), TV (**middle**), and HHBM (**right**) methods.

**Table 1 entropy-20-00977-t001:** RMSE, ISNR, PSNR, and SSIM of the reconstructed phantom with 50 global iterations (10 gradient descent iterations in each global iteration). The values of the regularization parameters are respectively $\lambda_{QR}=10$ and $\lambda_{TV}=50$ for SNR = 40 dB, $\lambda_{QR}=600$ and $\lambda_{TV}=100$ for SNR = 20 dB. TV, Total Variation; HHBM, Hierarchical Haar transform-based Bayesian Method; QR, Quadratic Regularization.

	256×256×256 **Shepp–Logan Phantom**											
	**180 Projections**						**90 Projections**					
	**40 dB**			**20 dB**			**40 dB**			**20 dB**		
	**QR**	**TV**	**HHBM**	**QR**	**TV**	**HHBM**	**QR**	**TV**	**HHBM**	**QR**	**TV**	**HHBM**
RMSE	0.0236	0.0114	0.0069	0.1309	0.0209	0.0755	0.0401	0.0212	0.0092	0.1558	0.0491	0.1117
ISNR	5.5584	8.7217	10.9346	7.2024	15.1775	10.2162	6.6136	9.3832	12.9973	8.4583	13.4765	9.9056
PSNR	30.0675	33.2308	35.4437	22.6318	30.6069	25.0209	27.7743	30.5439	34.1579	21.8754	26.8937	23.3227
SSIM	0.9999	0.9999	1.0000	0.9992	0.9999	0.9995	0.9997	0.9999	0.9999	0.9990	0.9997	0.9993
	**60 Projections**						**45 Projections**					
	**40 dB**			**20 dB**			**40 dB**			**20 dB**		
	**QR**	**TV**	**HHBM**	**QR**	**TV**	**HHBM**	**QR**	**TV**	**HHBM**	**QR**	**TV**	**HHBM**
RMSE	0.0636	0.0321	0.0107	0.1656	0.0753	0.1293	0.0904	0.0474	0.0132	0.1854	0.0901	0.1414
ISNR	9.3826	12.3480	17.1346	9.1492	12.5701	10.2226	10.3301	13.1308	18.6839	10.0137	13.1476	11.1916
PSNR	25.7693	28.7347	33.5214	21.6116	25.0325	22.6849	24.2404	27.0412	32.5942	21.1195	24.2535	22.2974
SSIM	0.9996	0.9995	0.9999	0.9990	0.9995	0.9992	0.9994	0.9997	0.9999	0.9988	0.9994	0.9991
	**36 Projections**						**18 Projections**					
	**40 dB**			**20 dB**			**40 dB**			**20 dB**		
	**QR**	**TV**	**HHBM**	**QR**	**TV**	**HHBM**	**QR**	**TV**	**HHBM**	**QR**	**TV**	**HHBM**
RMSE	0.1177	0.0680	0.0169	0.1957	0.1116	0.1500	0.2581	0.2104	0.0574	0.2907	0.2313	0.2014
ISNR	10.6591	13.0424	19.0933	10.8633	13.3032	12.0187	10.7122	11.5992	17.2373	10.8088	11.8022	12.4036
PSNR	23.0949	25.4783	31.5292	20.8865	23.3264	22.0420	19.6263	20.5133	26.1514	19.1085	20.1020	20.7033
SSIM	0.9993	0.9996	0.9999	0.9988	0.9993	0.9990	0.9983	0.9987	0.9996	0.9981	0.9985	0.9987

**Table 2 entropy-20-00977-t002:** The RMSE of the reconstructed $256^{3}$ Shepp–Logan phantom by using projection obtained from a $1024^{3}$ Shepp–Logan phantom.

RMSE	QR	TV	HHBM
180 proj	0.0581	0.0540	0.0558
90 proj	0.0655	0.0573	0.0610
60 proj	0.0846	0.0675	0.0690
45 proj	0.1079	0.0830	0.0783
36 proj	0.1326	0.1027	0.0882

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
