# Peer review of "Bayesian 3D X-ray Computed Tomography with a Hierarchical Prior Model for Sparsity in Haar Transform Domain"

_entropy, 2018, doi:10.3390/e20120977_

Reviewer 1 Report

The reviewed manuscript is related to the development of the hierarchical model for X-ray CT reconstruction. For this purpose the Haar transformation and and appropriate Bayesian computational methods were used. The manuscript is well written. It contains of large theoretical part and also an example. The methodology is well described.

Author Response

Thank you very much for the revision

Reviewer 2 Report

Overall, the paper is very interesting.

Some major points that should be addressed:

-- Were the initial simulations done using the same discretisation as the reconstruction? i.e. on a 256 cubed grid? If so, the results are not as impressive as they could be. To avoid this inverse crime (see, e.g. G. T. Herman https://doi.org/10.1088/0266-5611/24/4/045011) one can do the initial simulation at a high resolution and then downsample the results in projection space.

-- For the simulations only the analytical projection operator H f and the back-projection operator H T g are used. Considering that the costly projection and back-projection operators are computed in every iteration, the GPU processor is used via the ASTRA toolbox [63] to accelerate the computation.

Could the authors clarify what they mean by “analytical” here? My understanding is that the ASTRA toolbox does projection by numerical raytracing, which is not what I would understand as “analytical”. Is this perhaps referring to the inverse crime I mentioned above?

-- It would be good to be able to a visual comparison (e.g. as per figure 8 ) for increasing levels of noise.

Some minor points that should be addressed:

-- Reconstruction is expected to be achieved with fewer projections, which makes it an ill-posed inverse problem.

In point of fact, it already is ill-posed in the sense of Hadamard due to the non-continuous nature of the inverse operator (see Mathematics of computerised tomography, by Frank Natterer). Incomplete projection data also introduces a null space into the forward problem, further confounding the ill-posedness.

-- Iterative methods are used to optimize the reconstruction results.

A slight expansion on this would be appreciated (optimised subject to?).

-- In the regularization optimization, by taking into consideration the big data size in X-ray CT, the optimization algorithms such as gradient descent, conjugate gradient, can be used.

Is the big data size the thing that makes conjugate gradient work? If not, this sentence should be slightly re-worked to be clearer.

-- Sometimes the criterion of minimization consists of more than one regularization term

I didn’t understand this sentence. Perhaps this could be clarified/expanded slightly?

-- From the probabilistic point of view, a Gaussian model for the additive noise in the forward model

Should note that the poisson noise model also applies to X-ray CT detection (i.e. the physical imaging process, it’s not what’s assumed in most algorithms), and more generally any detection case with a fixed probability-of-detection-per-quanta of energy.

-- In industrial applications, especially for the NDT of a large-sized object, the size of the dataset becomes critical, and so does the projection and back-projection operators in CT.

“Critical” needs to be expanded on here, for the benefit of the uninitiated reader: what are the particular consequences of large data set sizes (e.g. prohibitively high computational load that scales as order N^4)?

Author Response

Dear reviewer,

Thank you very much for the review of the paper. We have corrected the paper according to the review and added the subsection 3.4 in the revision to compare the results of different methods when a different forward model is applied to the generation of projection and reconstruction.

Here we attached a response file.

Thank you!

Reviewer 3 Report

While my background is in X-ray computed tomography, I am not familiar with the mathematical aspects of tomographic reconstruction. I therefore cannot judge the scientific soundness of the mathematical derivations presented in the introductory section. However, the results presented support the claims of robustness of the novel method. I am attaching a word document in which I include my feedback on the grammar and writing style in 'Track Changes' mode. The formatting of the word document will have significant discrepancies from the PDF file submitted by the authors. This formatting issue is due to the conversion from PDF to DOCX. I hope my comments can be constructive despite my lack of expertise in the particular subject matter presented in this paper.

Author Response

Dear reviewer,

Thank you very much for the revision. We have corrected the paper and rewrite some paragraphs
according to the revision.

Thank you!